# Gene Copy Number Dictates Extracellular Vesicle Cargo

**DOI:** 10.3390/ijms26125496

**Published:** 2025-06-08

**Authors:** Sumeet Poudel, Zhiyong He, Jerilyn Izac, Lili Wang

**Affiliations:** National Institute of Standards and Technology (NIST), Gaithersburg, MD 20817, USA

**Keywords:** EV, EV cargo, EV protein loading, EV RNA loading, EV Imaging Flow Cytometry Analysis, EV biology

## Abstract

Extracellular vesicles (EVs) are membrane-surrounded vesicles that carry heterogeneous cellular components, including proteins, nucleic acids, lipids, and metabolites. EVs’ intravesicular and surface contents possess many biomarkers of physiological and pathological importance. Because of the heterogeneous cargo, EVs can mediate local and distal cell–cell communication. However, the way in which the genome signature regulates EV cargo has not been well studied. This study aimed to understand how genetics impact EV cargo loading. EVs were isolated from vector copy number cells with a fluorescent reporter (*GFP*) with varying inserted transgene copies and from NIST SRM 2373 cells (MDA-MB-231, MDA-MB-453, SK-BR-3, and BT-474), which contain varying copies of the *HER2* gene. Spectradyne nCS1 was utilized to count EVs and measure size distribution. Imaging Flow Cytometry was used to analyze the surface protein content of single EVs and for total EV counts. The RNA content of the EVs was measured using ddPCR. Our results from stable reporter cell lines and breast cancer cell lines suggest that the gene copy number dictates the protein cargo of the EVs but not the RNA content. Increasing copies of a reporter gene (*GFP*) or a naturally occurring gene (*HER2*) from breast cancer cells correlated with increasing EV counts positive for the protein cargo compared to total EV counts until a copy threshold was reached. This study has broad implications for understanding EV biology in the context of cancer biology, diagnostics, EV biology/manufacturing, and therapeutic delivery.

## 1. Introduction

Extracellular vesicles (EVs) are lipid bilayer-enclosed particles secreted by cells into the extracellular space, playing a crucial role in intercellular communication. These vesicles, which include exosomes (30–150 nm), microvesicles (100–1000 nm), and apoptotic bodies (>1000 nm), are produced by diverse cell types and can be found in various biological fluids, such as blood, urine, cerebrospinal fluid, and saliva [1,2]. EVs are typically rich in tetraspanin (TS) proteins, including CD9, CD63, and CD81. In addition, they contain other key proteins, such as ALIX, which plays a role in cellular regulation; TSG101, which is associated with tumor susceptibility; MHC1, which is involved in immune response; and HSP90, a heat shock protein crucial for protein stabilization [2,3]. Once considered cellular debris, EVs are recognized as active biological particles capable of transferring functional biomolecules between cells [4]. The molecular cargo carried by EVs—comprising proteins, lipids, nucleic acids, and metabolites—reflects the physiological state of their cells of origin and plays a pivotal role in mediating biological processes [5,6]. EV cargo has been implicated in numerous functions, including immune modulation, tissue repair, and cellular signaling [7,8]. Furthermore, EVs can facilitate the horizontal transfer of genetic information through their RNA and DNA content, influencing gene expression in recipient cells [9]. The ability of EVs to transport bioactive molecules in a stable, protected manner across biological barriers has also positioned EVs as promising candidates for therapeutic delivery systems and as diagnostic biomarkers [10]. Recent studies have demonstrated their potential utility in cancer, neurodegenerative diseases, and cardiovascular disorders [11,12].

EV cargo loading into vesicles is a highly regulated and selective process. Exosomes, for example, originate from the inward budding of endosomal membranes to form multivesicular bodies, where specific cargoes such as TS, miRNAs, and ubiquitinated proteins are incorporated. This process is mediated by mechanisms involving the endosomal sorting complexes required for transport (ESCRT) machinery, lipid raft domains, or ESCRT-independent pathways [13,14]. Similarly, microvesicle cargo is packaged during the outward budding of the plasma membrane, influenced by cytoskeletal reorganization and signaling molecules such as small GTPases [3]. However, understanding the mechanisms underlying the selective packaging of EV cargo remains a major scientific challenge [15,16]. In this study, we attempt to shed some light on the selective loading of EV cargo from a genetic viewpoint.

We explored the impact of genetic copy number variation on EV cargo loading. EVs from engineered cells with various vector copy numbers (VCNs) between 0 and 4 of a green fluorescent protein (*GFP*) reporter gene [17], as well as from National Institute of Standards and Technology (NIST) Standard Reference Material (SRM) 2373 reference cells (MDA-MB-231, MDA-MB-453, SK-BR-3, and BT-474) [18], which contain different human epidermal growth factor receptor 2 (*HER2*) gene copy numbers, were used to show whether the gene copy number plays a role in determining the cargo of EVs. SRM 2373 cell lines used in this study were certified with *HER2* amplification ratios that range from approximately 1 to 17 copies relative to selected reference genes (*DCK*, *EIF5B*, *RPS27A*, and *PMM1*). The ratios are 1.3, 2.9, 9.7, and 17.7 for MDA-MB-231, MDA-MB-453, SK-BR-3, and BT-474, respectively, relative to reference genes [19]. This study aimed to determine whether the gene copy number influences the production of EVs carrying gene-derived protein or RNA and to assess whether cell lines with varying gene copy numbers can help identify correlations between gene copies and the abundance of such EVs.

## 2. Results and Discussion

### 2.1. Microfluidic Resistive Pulse Sensing (MRPS) Analysis of Copy Number Cell-Derived EVs Shows Variability in Count and Size

Serial dilutions of VCN and SRM 2373 cell-derived EVs were measured by MRPS using C-400 cartridges, which can measure particles of 65 nm to 400 nm in diameter. The samples were diluted from the stock EV suspensions with filtered Phosphate Buffer Saline (PBS) containing 1% (*v*/*v*) Tween 20 to 5 × 10^8^ particles/mL and 5 × 10^9^ particles/mL. EVs stored in 1% Tween20/PBS at 4 °C for 48 h exhibited minimal degradation while effectively retaining their size and structure [20,21]. VCN cell-derived EV counts ranged from ~2 × 10^10^ to 6 × 10^10^ particles/mL (Figure 1A, Table 1), while SRM 2373 cell-derived EV counts ranged from ~9 × 10^9^ to 4 × 10^10^ particles/mL (Figure 1B, Table 2). Figure 1C shows a typical nCS1 Data Analyzer Software output with analysis and measurement parameters, such as the size range of the cartridges, concentrations, statistical errors, volume intake, time, etc., for a representative diluted VCN4 cell EV sample. We observed a particle size distribution of EV samples ranging from 65 nm to 350 nm, with a weighted mean diameter of approximately 90 nm (Figure 1D, representative data for VCN4 EV shown). The distribution showed the highest frequency counts in the lower diameter range (65–85 nm), with a gradual decrease in frequency as the particle diameter increased. The particle size distribution approximates a normal distribution, with rightward skew at higher diameters and a clear preference for the formation of smaller particles under acquisition conditions (Figure 1D). We noticed a large variability in the concentration and size measurement (Figure 1, Table 1 and Table 2) due to the inherent heterogeneity of the EV samples and the challenge posed by the nanoscale of particle size measurement [22,23]. Variability in size and concentration was observed both within and between the sets of EV study samples (VCN EVs and SRM 2373 EVs). This variability in concentration (Table 1 and Table 2) and size persisted throughout various extractions despite timing media collection for EV extraction at approximately 80% cell confluence. Factor-like differences in cell population doubling times between the two cell sets (VCN and SRM 2373) likely contributed to this variation [24]. In addition, the cell culture conditions between the two sets (i.e., adherent (SRM 2373) vs. suspension (VCN cells)) likely impacted the EV concentrations [24] as adherent cell growth is limited by the surface area of the flask, while suspension cells rely on the volume of the media. We also noticed such variabilities among the EV extracted from the same sets (VCN or SRM 2373).

### 2.2. Stable GFP VCN Cell-Derived EVs Reveal Correlation with Protein Cargo but Not RNA Cargo

VCN cells contain constitutively expressed, stably integrated *GFP* gene within the Jurkat genome. They contain 1–4 copies of *GFP* inserted at uncontrolled genomic loci. Cells containing 1 copy of GFP are called VCN 1, those with 2 copies are called VCN 2, and so on. GFP intensity in VCN cell lines correlates with the increasing copy number of the lentiviral vector integrated reference gene (*GFP*), consistent with previous reports (Figure 2A [17]). Representative single-cell imaging by Imaging Flow Cytometry (IFC) shows that VCN 0 cells lack GFP expression (Figure 2B,C), whereas VCN 4 cells exhibit the highest GFP expression. IFC analysis confirmed that higher *GFP* copy numbers correspond to increased GFP intensity in VCN cells (VCN 0 vs. VCN 1, 2, 3, and 4; Figure 2A,D), and GFP intensities are consistent with the genomic vector copy number for each evaluated cell line in the VCN cell series.

EVs were isolated from VCN cell lines and characterized using IFC after staining with a TS Allophycocyanin (APC) antibody. As described in the Methods Section under data analysis in the Image Data Exploration and Analysis Software (IDEAS), our gating strategy first distinguished speed beads (SBs) from other particles (not SBs), designating them “not SBs” population for intensity-based analysis. A dot plot analysis of the “not SBs” population identified two distinct groups: one consisting of only GFP^+^ particles and another comprising TS^+^/GFP^+^ particles (Figure 3A, left). The GFP^+^-only particles are likely protein aggregates, autofluorescent particles, TS-negative EVs, or contaminants detected in the GFP channel. Imaging confirmed that the GFP^+^-only population (Figure 3B, bottom) lacked TS APC staining, whereas the TS^+^/GFP^+^ population (Figure 3B, top) was positive for both GFP and TS APC. The overlay images further support that GFP^+^ EVs align with TS APC^+^ staining (Figure 3B, top), confirming their identity as EVs, unlike the GFP^+^ particles, which are likely TS-negative EVs or non-EV contaminants (Figure 3B, bottom).

For further analysis, we focused on the TS^+^/GFP^+^ EV population using intensity-based histograms to quantify TS APC^+^ EVs (Figure 3A, center) and GFP^+^ EVs (Figure 3A, right). The control samples for VCN cell-derived EV cargo IFC analysis using representative VCN4-derived EV samples, including buffer only, antibody only, single-stained EVs, and unstained EVs, are presented in Appendix A. The counts of TS APC^+^ and GFP^+^ EVs from each VCN cell line were used to determine the proportion of GFP^+^ EVs relative to the total EV population (TS^+^) and their corresponding percentage positivity (Figure 3C, Table 3). As the *GFP* copy number increased in VCN cell lines, a corresponding rise in the GFP^+^ EV population was observed (Figure 3C, Table 3). Specifically, VCN4-derived EVs (containing four copies of *GFP*) exhibited approximately 74% GFP positivity, while VCN3 EVs showed ~57%, VCN2 EVs showed ~31%, and VCN1 EVs showed ~23% GFP positivity. As expected, the VCN0 cell-derived EVs displayed 0% GFP positivity. These results show that the EV (GFP) protein cargo is directly correlated with gene (*GFP*) copy numbers.

Next, we examined the relative RNA expression in EVs to the *GFP* gene copy number. An analysis of RNA expression from VCN 0–4 cells revealed a progressive increase in *GFP* RNA expression relative to the reference *GAPDH* RNA (Figure 3D). However, this correlation was not observed in GFP^+^ EVs. Unlike protein cargo, *GFP* RNA expression did not scale proportionally with the gene copy number. Specifically, after normalization to VCN1 EV relative to the *GFP* RNA expression level, VCN2 exhibited a similar relative *GFP* level as VCN1, while VCN3 and VCN4 showed approximately 2-folds the expression of VCN1 instead of 3- or 4-folds. As expected, VCN0 displayed no *GFP* expression in both the cellular and EV analyses (Figure 3D). While the cellular *GFP* RNA levels increased with the gene copy number, a correlation with **RNA cargo in EVs** was not observed. Taken together, this suggests that GFP protein is efficiently translated and incorporated into EVs, but *GFP* RNA loading into EVs does not follow the same trend.

### 2.3. SRM 2373 Cell-Derived EVs Exhibit a Partial Correlation with Protein Cargo in the Context of HER2 Genes but Not with RNA Cargo

*HER2*, officially known as *ERBB2*, is a proto-oncogene that encodes a transmembrane glycoprotein with tyrosine kinase activity [25]. *HER2* gene amplification leads to increased protein expression, a phenomenon observed in roughly 20% of breast cancers, and is linked to poor patient outcomes [26,27]. SRM 2373 is composed of genomic DNA extracted from five distinct breast cancer cell lines, each exhibiting varying levels of *HER2* gene amplification. In our study, we utilized four of these cell lines—MDA-MB-231, MDA-MB-453, SK-BR-3, and BT-474, which have certified *HER2* gene copy number ratios of 1.3, 2.9, 9.7, and 17.7, respectively, relative to the selected reference genes [19]. MDA-MB-231 cells have low HER2 protein expression and are considered HER2-negative, while MDA-MB-453, SK-BR-3, and BT-474 express high HER2 levels and are considered HER2-positive [28,29]. An intensity-based analysis of SRM 2373 cells using IFC confirmed that MDA-MB-231 cells have low HER2 expression, while MDA-MB-453, SKBR3, and BT-474 exhibit high HER2 levels (Figure 4A). Representative single-cell imaging analyses by IFC confirmed these observations (Figure 4B). Although MDA-MB-453 is classified as a high HER2 expression cell line, SK-BR-3 and BT-474 express HER2 at significantly higher levels, with log-fold increases compared to MDA-MB-453 (Figure 4A,C). Despite BT-474 cells having more *HER2* gene copies than SK-BR-3 cells, BT-474 cells show similar HER2 protein expression to SK-BR-3 in our analysis (Figure 4A,C). Previous reports indicate similar or lower HER2 expression in BT-474 cells compared to SK-BR-3 cells [28,29] likely due to factors like transcriptional regulation, mRNA stability, or post-translational modifications, which can affect protein expression efficiency [28,29,30].

EVs were isolated from SRM 2373 cells and characterized using IFC after staining with a TS Phycoerythrin (PE; EV^+^) and HER2-AF647 antibodies. The gating strategy previously described for VCN-cell EV was also applied to SRM 2373-cell EVs to remove SBs from the population. Then, a dot plot analysis of the “not SBs” population was performed to identify two distinct groups: one consisting of only HER2^+^ particles and another comprising TS PE^+^/HER2 AF647^+^ EVs (Figure 5A, left). The HER2^+^-only particles are likely non-EVs, protein aggregates, autofluorescent particles, TS-negative EVs, or contaminants detected in the HER2 channel. Imaging confirmed that the HER2^+^-only population (Figure 5B, bottom) lacked TS PE staining, whereas the TS PE^+^/HER2 AF647^+^ population (Figure 5B, top) was positive for both HER2 and TS PE. The overlay images further support that HER2^+^ EVs align with TS PE^+^ staining (Figure 5B, top), confirming their identity as EVs, unlike the HER2^+^-only particles, which are likely TS-negative EVs or non-EV contaminants (Figure 5B, bottom).

For further analysis, we focused on the TS PE^+^/HER2 AF647^+^ EV population using intensity-based histograms to quantify TS PE^+^ EVs (Figure 5A, center) and HER2 AF647^+^ EVs (Figure 5A, right). The control samples for SRM2373 cell-derived EV cargo IFC analysis using representative BT-474-derived EV samples, including buffer only, single or dual antibody only, EVs stained with full minus one (FMO), and unstained EVs, are presented in Appendix A. The counts of TS PE^+^ and HER2 AF647^+^ EVs from each SRM 2373 cell were used to determine the proportion of HER2^+^ EVs relative to the total EV population (TS PE^+^) and their corresponding percentage positivity (Figure 5C, Table 4). In SRM 2373 cell EVs, an increase in the HER2 gene copy number correlated with a rise in the HER2^+^ EV population (Figure 5C, Table 4) until reaching a threshold where HER2^+^ EV levels decreased. Specifically, BT-474-derived EVs, which contain ~17.7 *HER2* gene copies relative to reference gene, exhibited approximately 69% HER2 positivity. In comparison, SK-BR-3 EVs (~9.7 copies) showed ~73% positivity, MDA-MB-453 EVs (~2.9 copies) showed ~33%, and MDA-MB-231 EVs (~1.3 copies) showed ~6% positivity, suggesting that the threshold was met at ~9 copies (SK-BR-3 cells) in our analysis. These findings indicate a direct correlation between the *HER2* gene copy number and HER2 protein levels in SRM 2373 EVs up to a certain threshold for gene copy numbers. However, it is important to note that this threshold may vary with different cell lines and genetic loci, necessitating further analysis on a case-by-case basis.

Next, we examined RNA relative expression in the EVs of the SRM 2373 cells in the context of the *HER2* gene copy number. An analysis of *HER2* RNA from SRM 2373 cells revealed an increase in relative *HER2* RNA expression and a positive correlation with the *HER2* gene copy number (Figure 5D). However, this correlation was not observed in the relative *HER2* RNA expression of the EVs; this parallels observations from the VCN cell-derived EVs (Figure 3D). Unlike protein cargo, *HER2* RNA expression did not scale proportionally with the gene copy number. Specifically, after normalization to MDA-MB-231 *HER2* EV RNA expression, MDA-MB-453, SK-BR-3, and BT-474 showed relative *HER2* RNA expression that was 20-, 12-, and 15-fold compared to that of MDA-MB-231, respectively (Figure 5D). While the cellular *HER2* RNA levels increased proportionally with the gene copy number, the correlation with RNA cargo in EVs did not. Taken together, this suggests that HER2 protein is efficiently translated and incorporated into EVs based on the copy number up to a certain gene copy number threshold until a decrease or plateau occurs, but *HER2* RNA loading into EVs does not follow the same trend.

## 3. Materials and Methods

### 3.1. Cell Culture

The generation and maintenance of VCN cell series have been previously described [17]. In brief, a parental Jurkat cell line (VCN 0) and VCN reference standard clonal Jurkat cell lines (VCN 1, 2, 3, and 4) were cultured in RPMI-1640 growth medium (ATCC Cat # 30-2001, Manassas, VA, USA) supplemented with 10% heat-inactivated fetal bovine serum (FBS, Hyclone, Logan, UT, USA, Cat # SH30071.03HI) and two mM LGlutamax (Gibco™ GlutaMAX™ Supplement, Grand Island, NY, USA, Cat # 35050061). Cultures were maintained at a cell concentration between 2 × 10^5^ and 1 × 10^6^ cells/mL with fresh medium.

Detailed information on the culture of the four cells from the NIST SRM 2373 has been previously described [19]. Briefly, MDA-MB-231 and MDA-MB-453 cells were cultured in Leibovitz’s L-15 Medium (ATCC # 30-2008) supplemented with 10% FBS (Gibco # 10437-028) at 37 °C in an air atmosphere without added CO_2_. SK-BR-3 cells were grown in McCoy’s 5A modified medium (ATCC # 30-2007) supplemented with 10% FBS at 37 °C in a humidified 5% CO_2_, 95% air atmosphere. BT-474 cells were cultured in Hybri-Care Medium (ATCC # 46-X) supplemented with 1.5 g/L sodium bicarbonate and 10% FBS at 37 °C in a humidified 5% CO_2_, 95% air atmosphere.

Cultured cells described above were grown in respective media containing 10% EV-depleted FBS (fetal bovine serum, exosome-depleted, Thermo Fisher, Millersburg, PA, USA, Cat #A2720801) for 48 h before EV extraction.

### 3.2. EV Extraction

The EVs from cell culture media cultured with EV-depleted FBS were extracted with an Invitrogen™ Total Exosome Isolation Reagent (from cell culture media) kit (Carlsbad, CA, USA, Cat # 4478359) following the manufacturer’s protocol.

### 3.3. Microfluidic Resistive Pulse Sensing (MRPS)

MRPS measurements were collected using a Spectradyne nCS1 instrument (hardware version 2, Signal Hill, CA, USA). C-400 (65 nm to 400 nm diameter size range) cartridges were used for EV measurements. Spectradyne cartridges are pre-calibrated by the manufacturer for size and number concentration. Filtered (200 nm) PBS with 1% (*v*/*v*) polysorbate 20 (Tween^®^20, Sigma-Aldrich, Burlington, MA, USA) was used as a running buffer in the post-nanoconstriction flow channel to enable appropriate flow and washing. Several dilutions (1×, 5×, and 10×) of each EV sample were measured by diluting the stock suspensions with 20 nm pore-filtered (Cytvia Inorganic Membrane Filter 0.02 µm, Marlborough, MA, USA, Product # 6809-2002) PBS. Collected data were analyzed using nCS1 Data Analyzer Software (version: 2.5.0.325). Cumulative concentration measurements from the three extractions were used to calculate the mean and standard deviation. The particle size distribution analysis was performed to characterize the dimensional properties of the produced EVs. The raw data collection encompassed diameters ranging from 65 to 400 nm (limit of C-400 cartridges), with their corresponding frequency distributions. A histogram analysis was used to visualize the size distribution pattern to characterize the population parameters. Frequency-weighted calculations with varying population densities across different size ranges were used for the distribution analysis.

### 3.4. EV Staining for Imaging Flow Cytometry (IFC)

Twenty microliters of approximately 1 × 10^8^ particles/microliter of EVs (based on the Spectradyne nCS1 counts) were used as starting materials for staining. For EVs derived from VCN cells (0–4), 2 µL of Anti-Human Tetraspanin vTAG™ Antibody Cocktail PE (Cellarcus Biosciences, La Jolla, CA, USA, SKU # CBS5-PE-100T) was added, mixed by pipetting, and incubated for 1 h in the dark at room temperature. For EVs derived from SRM 2373 cells (MDA-MB-231, MDA-MB-453, SK-BR-3, and BT-474), 2 µL of Anti-Human Tetraspanin vTAG™ Antibody Cocktail APC (Cellarcus Biosciences, SKU # CBS5-APC-100T) and 1 µL of AF647 anti-human CD340 (erbB2/HER-2) Antibody (BioLegend, San Diego, CA, USA, Cat # 324412) were added and mixed, and the mixture was incubated for 1 h in the dark at room temperature.

### 3.5. IFC Measurements

Assay controls for IFC were incorporated into all experiments following the guidelines of the MIFlowCyt-EV framework [31]. These controls included filtered PBS, filtered PBS with antibodies, unstained EV samples, single-stained samples, isotype controls matched with their respective fluorophore-conjugated monoclonal antibodies at identical concentrations, FMO, and samples treated with detergent. For the detergent treatment, 10% (*v*/*v*) Triton X-100 stock solution was used. Specifically, 20 µL of the Triton X-100 stock solution was added to achieve a final concentration of 0.5% (*v*/*v*) per test. The samples were then incubated at room temperature in the dark for 30 min before acquisition.

All samples were analyzed on a Cytek^®^ Amnis^®^ ImageStream^®X^ (ISX) MKII IFC (Seattle, WA, USA) instrument equipped with five lasers (405 nm, 488 nm, 561nm, 642 nm, and 785 nm (side scatter (SSC)) by diluting the stained EVs 10-fold. The following lasers (and powers) were utilized for this study: 488 nm (300 mW), 642 nm (150 mW), and 785 nm (1.33 mW). The 405 nm laser, 561 nm laser, and BF imaging was turned off for EV analysis. The instrument calibration tool ASSIST^®^ was used upon each startup to ensure the comparability of measurements. The ISX is equipped with three objectives (20×/40×/60×) and 1 CCD camera. For EV analysis, all data were acquired using the 60× objective (numerical aperture of 0.9—wherein one image pixel has an area of 0.1 µm^2^) with the high gain (HG) checked and fluidics settings set to “low speed/high sensitivity”—resulting in a flow speed of 43.59 ± 0.07 mm/s (mean ± standard deviation). Data were acquired over 120 seconds for standardization after the fluidics stabilized. GFP fluorescence signals were collected in channel 2 (505–560 nm filter), PE signals in channel 3 (560–595 nm filter), and APC/AF 647 signals in channel 5 (642–745 nm filter). Channel 6 (745–785 nm filter) was used for SSC detection. Particle enumeration was achieved through the advanced fluidic control of the ISX coupled with continuously running SBs, used by the IFC to measure sample velocity for camera synchronization during acquisition, enabling particle enumeration during analysis and the application of the “objects/mL” feature within the ISX IDEAS^®^ version 6.4. Likewise, for cellular analysis, BF in channel 4, GFP in channel 2, and AF647 in channel 5 were used with the following lasers and powers: 488 nm (20 mW), 642 nm (50 mW), and 785 nm (1.33 mW) at a 60× objective.

### 3.6. Data Analysis with IDEAS Software (Version 6.3)

Data analysis was conducted using the IDEAS software. The image display settings were linearly adjusted for all fluorescent events within each channel and consistently applied across all files from the respective experiments. The IDEAS software employs “masks” (M), which are algorithms that select pixels within an image based on their intensity and localization to define the analysis area for each event within the pixel grid. The “masks combined” (MC) standard setting was used to measure all fluorescence intensities in the channels corresponding to the fluorochromes (Ch02 for VCN cells; Ch05 for NIST SRM 2373 HER2 cells).

We used the following gating strategy for cellular analysis: we created a dot plot of the area of the BF channel vs. the aspect ratio of the BF channel on all particles to gate for single cells (area >100 and <1000 square microns; aspect ratio >0.8 and <1), and then the single cell populations were sorted by Gradient RMS of BF to obtain focused cells (>50 intensity units), then intensity (Intensity_MC_Ch02 or Ch05) of specific channels (e.g., GFP or HER2 AF647) of the focused cells for reporting.

For EV intensity analysis, the “mask” (M) that corresponds to specific channel fluorescence corresponding to the fluorochrome was used. The following gating strategy was used for the analysis of EVs: The raw MaxPixel of channel 6 (SSC) was used to separate “SB” and “not SB” events, and then an intensity-based dot plot was created for two fluorescent channels of interest (TS APC^+^ vs. GFP for VCN EVs or TS PE^+^ vs. HER2 AF647^+^ for SRM 2373 EVs). Then, an intensity-based histogram was created for the TS^+^ population (APC^+^ for VCN EVs or PE^+^ for NIST SRM 2373 EVs), and finally, another intensity-based histogram was created for the protein of interest (GFP^+^ or HER2 AF647^+^) from the TS^+^ population. A single positive population of GFP^+^ or HER2^+^, which did not stain for TS (TS^−^), in the dot plots was considered a cellular protein aggregate, non-EV, TS-negative EV, or antibody aggregate.

Fluorescent events from singly stained samples were utilized to set compensation matrices, addressing the spectral overlap between fluorochromes. Single-positive gating areas were established based on the single-positive fluorescent populations, and double-positive gates were determined based on the boundaries of the single-positive gates. Unstained samples helped define the low end of the various gates.

### 3.7. RNA Extraction

An amount of 50 µL of approximately 1 × 10^9^/µL EVs (based on the Spectradyne nCS1 counts) was used as starting material for EV RNA extraction, and approximately 2 × 10^5^ cells were used for cellular RNA extraction. The total cellular RNA was extracted using the QIAGEN RNeasy Plus Mini Kit (Germantown, MD, USA, Cat # 74134) by following the manufacturer’s protocol. EV RNA was extracted by using the Invitrogen™ Total Exosome RNA & Protein Isolation Kit (Carlsbad, CA, USA, Cat # 4478545) by following the manufacturer’s protocol. Three replicate extractions were performed using media from 3 different passages.

### 3.8. Relative RNA Quantification by Digital Droplet Polymerase Chain Reaction (ddPCR)

The total RNA isolated from breast cancer cells and EVs was diluted in RNA storage solution (RSS, Thermo Fisher, Cat # AM7000) with 20 ng/µL yeast tRNA (Thermo Fisher, Cat # AM7119) to a final concentration of 60–150 pg/µL. Diluted RNA (total) was used for the determination of relative *GFP* with TaqMan™ minor groove binder (MGB) probe sets [32] or *HER2* by using the One-Step RT-ddPCR Advanced Kit (Bio-Rad, Hercules, CA, USA, Cat # 1864022). *HER2*-specific primers and probe sequences were published before [19]. *GAPDH* expression assay (including primers and probe) was purchased from Thermo Fisher (Cat # 402869) and was used as an internal control. The ddPCR reactions were set up by following the kit protocols. After the droplets were generated on the Bio-Rad manual droplet generator, they were transferred to a Bio-Rad 96-well ddPCR plate, followed by thermal cycling amplification. The thermal cycling steps were 25 °C for 3 min for 1 cycle; 50 °C for 60 min for 1 cycle; 95 °C for 10 min for 1 cycle; 95 °C for 30 s and 58 °C for 60 s for 40 cycles; 98 °C for 10 min for 1 cycle; and then storage at 4 °C. The positive droplets were determined on a Bio-Rad QX200 droplet reader. The relative RNA quantifications were normalized to the *GAPDH* level of each sample.

## 4. Conclusions

MRPS overestimated the EV counts by up to a log-order magnitude compared to the IFC analysis. While other studies have reported similar EV counts using MRPS and Flow Cytometry [23], both VCN- and SRM 2373-derived EVs showed MRPS particle concentrations of ~10^10^ particles/mL, whereas the IFC measurements were ~10^9^ particles/mL. This discrepancy is likely due to differences in sensitivity and detection capabilities between methods [25]. The IFC concentration measurements are EV-specific, relying on TS PE or TS APC antibody staining, whereas MRPS is non-specific and counts all particles, including non-EVs, large aggregates, and other contaminants. Additionally, if a subset of EVs either lacks detectable levels of TS markers or does not express them at all [33], IFC-based EV counts are affected, resulting in lower counts. Variability in MRPS concentration measurements may also result from the inherent heterogeneity of EV samples and the challenges associated with analyzing particle sizes smaller than 65 nm because of the cartridge used [22,23]. Additionally, differences in the particle size range detected by IFC and MRPS can further contribute to variability between instruments.

The gene copy number dictates the EV protein cargo but not the RNA cargo. To our knowledge, this is the first report that shows how copy number variation in genes regulates the protein content in EVs. In this study, we provide evidence that increasing copies of a reporter gene (stably engineered *GFP*) or naturally occurring genes (*HER2*) is correlated with an increasing protein cargo-positive EV population compared to total EV counts. It is interesting to note that VCN4 cells, which contain only four copies of the stably engineered gene, produced approximately 74% GFP^+^ EVs relative to the total EVs. In comparison, both SK-BR-3 and BT-474 cells from SRM 2373, which have significantly more copies of the *HER2* gene, generated around 70–75% HER2^+^ EVs. This difference is likely due to multiple interacting factors related to the cellular and genetic machinery and origins of EVs, including cell doubling times, culture conditions (adherent SRM 2373 cells vs. suspension VCN cells), cargo protein type (transmembrane protein (HER2) vs. ubiquitous protein (GFP)), gene promoter strength (strong *EF1α* promoter in VCN cells vs. *HER2* promoter in SRM 2373 cells), and loci type (amplified endogenous *HER2* locus vs. engineered uncontrolled insertion of *GFP* locus), among others. It is important to note that having access to advanced analytical methods like IFC is advantageous in capturing rare GFP /HER2-positive populations, which would have been difficult to distinguish using traditional methods, particularly for low-copy events. The current findings, which focus on two specific genes (*GFP* and *HER2*), should be evaluated for generalizability to other genes. We also acknowledge that the results of the precipitation-based method of EV isolation need to be followed up with using other traditional extraction methods (like size exclusion chromatography, tangential flow filtration, centrifugation, etc.), and further explorations of the potential causes underlying the observed protein threshold effects are needed. The advancement of EV-based therapies depends on continued research efforts to refine how the fate of EV cargo is regulated from genetic, transcriptional, and translational perspectives.

For high copy genes containing cells like SRM 2373, there was a correlation between *HER2* gene copies and HER2^+^ EVs until a threshold was met, and then a decrease in HER2^+^ EVs counts was observed. Beyond the gene copy number threshold, the HER2^+^ EV levels decreased. Specifically, BT-474-derived EVs, which contain ~17.7 *HER2* gene copies relative to the reference gene, exhibited approximately 69% HER2 positivity, while in comparison, SK-BR-3 EVs (~9.7 copies) showed ~73% positivity. These findings indicate cellular or EV cargo regulatory mechanisms.

Our study further found that the gene copy number may not proportionally regulate the RNA content of EVs, although it is correlated with protein cargo. This may seem counterintuitive given that the central dogma dictates that RNA serves as the template for protein synthesis. However, the relationship between the RNA copy number and protein expression is nonlinear, influenced by transcriptional, post-transcriptional, and translational regulation. mRNA levels alone often poorly predict protein abundance, with variability depending on the cell type, tissue, and gene-specific factors. While a correlation exists [34], its strength varies [35], highlighting additional regulatory influences. Moreover, our RNA analysis using ddPCR may not fully capture the complexity of EV RNA profiles, particularly due to RNA fragmentation, a stability issue frequently observed in EV RNA. Given that EVs carry diverse RNA species, including intact and fragmented forms [36], ddPCR’s ability to detect and quantify transcripts may be limited. Studies suggest that long RNAs within EVs may be partially degraded, while fragmented mRNAs can still contribute to protein synthesis, sometimes leading to functional differences from their source cells [36,37]. For instance, RNA-Seq analysis has revealed a large diversity of long RNA transcripts in EVs, highlighting the complexity of their RNA cargo [36]. Advanced methods such as RNA sequencing could provide a more comprehensive view of EV RNA cargo, shedding light on the integrity, diversity, and functional potential of these RNA molecules. In general, protein assays are direct and often more robust in low-input samples than RNA-based methods like RT-qPCR or ddPCR, which are more prone to technical variation and amplification biases, especially in degraded or low-abundance samples such as EVs. Further studies have also shown that extracellular RNAs, particularly exosomal miRNAs, are often heavily fragmented and low in abundance, resulting in variable detection across platforms and frequent underestimation by ddPCR, thus underscoring the importance of RNA-Seq for accurate and unbiased transcriptome profiling [38,39]. These studies indicate that both intact and fragmented RNA within EVs may have distinct biological roles in recipient cells, further challenging the straightforward expectation of RNA directly determining protein output [40]. Studies have demonstrated that EVs selectively package specific RNAs, such as *RNY3*, *vtRNA*, and *MIRLET-7*, along with associated proteins like YBX1, IGF2BP2, and SRSF1/2, indicating a regulated sorting mechanism distinct from their cellular origin [36,41]. An integrative multi-omics analysis of breast cancer cells revealed weak correlations between EV protein and RNA content and their corresponding levels in parental cells [42], suggesting that EVs selectively enrich certain molecules rather than merely mirroring the source cell’s molecular composition. Essentially, the mechanisms governing the packaging of RNA and proteins into EVs are distinct and likely highly regulated [13,14,43].

This study has significant implications across multiple areas of cancer biology, diagnostics, EV biology/manufacturing, and therapeutic delivery. In cancer diagnostics, particularly for early-stage cancers, counting EVs derived from oncogenic cells with altered gene copy numbers, such as blood, urine, or serum samples, may offer a viable alternative for assessing cancer progression and diagnosing the disease. The presence of biomarkers in EVs, even at low concentrations in early cancers [44], suggests that copy number studies of EVs could serve as valuable diagnostic tools. In the context of cancer biology, the relationship between the gene copy number and EV cargo raises important questions about how gene copy number alterations affect the EV content of other oncogenes, and understanding this dynamic could enhance our understanding of cancer progression. From a manufacturing perspective, our findings raise the possibility of using gene copy number variations to strategize EV production. Rather than relying on traditional methods that involve attaching a stable cDNA to a transfection system under a strong viral promoter, increasing gene copy numbers could provide an alternative approach for generating EVs with desired properties. This strategy could be combined with existing EV manufacturing platforms to optimize the production yield. Furthermore, in the realm of therapeutic delivery, this research could influence strategies for loading proteins of interest into EVs for targeted therapies. The ability to manipulate gene copy numbers may enable more efficient loading and delivery of therapeutics via EVs, potentially enhancing the efficacy of treatments. Advancements in single EV tracking and genetic analysis also offer valuable insights into potential biomarkers and genetic alterations that could be used for early cancer detection. These novel approaches provide clues for identifying key genetic features within EVs, helping refine diagnostic and therapeutic strategies. Finally, the integration of these findings with best practices from organizations like NIST could offer new methods for manufacturing EV references and standards, providing scalable and efficient platforms for clinical applications.

## Figures and Tables

**Figure 1 ijms-26-05496-f001:**
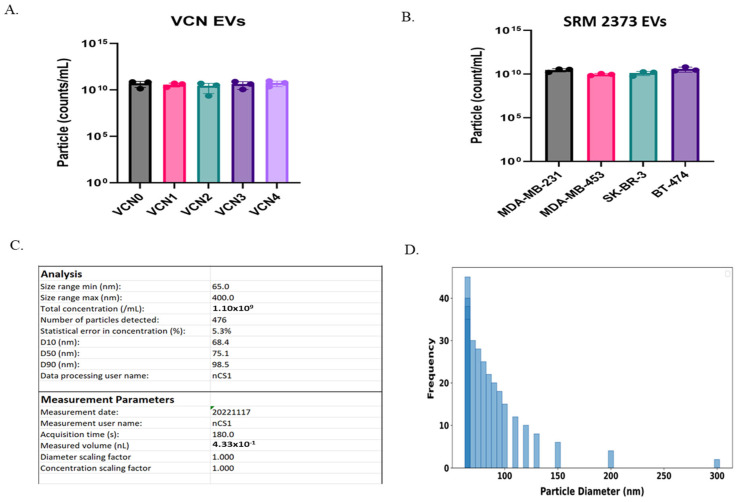
Analysis of VCN and SRM 2373 EVs by MRPS (Spectradyne nCS1). (**A**) Bar graph showing particle count of VCN 0–4 cell-derived EVs with Spectradyne nCS1; particle concentration (counts/mL) is shown along with their standard deviation from three extractions. (**B**) Bar graph showing particle count of NIST SRM 2373 cell-derived EVs with Spectradyne nCS1; particle concentration (counts/mL) is shown along with their standard deviation from three extractions. (**C**) Image of analysis and measurement parameters showing analysis details, such as size range of cartridges, concentrations, statistical errors, volume intake, time, etc., for single VCN4 EV acquisition from Spectradyne nCS1. (**D**) Particle size distribution profile of representative VCN4 EV analysis; frequency-weighted calculations with varying population densities across different size ranges were used for distribution fitting.

**Figure 2 ijms-26-05496-f002:**
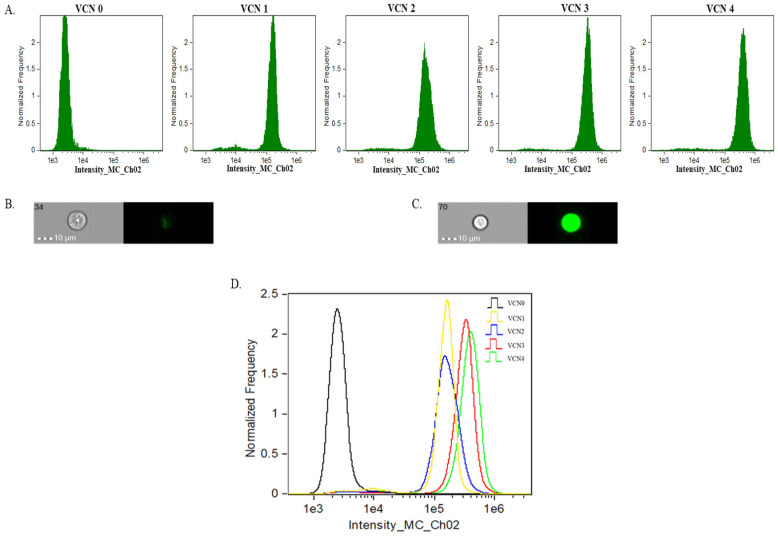
Analysis of VCN cell lines in IFC. (**A**) GFP intensity profile of focused single cells for VCN 0–4 cells (left to right). (**B**) Representative image of VCN 0 cells in GFP and brightfield (BF) channels. (**C**) Representative image of VCN 1–4 cells in GFP and BF channels. (**D**) Histogram overlay of (**A**); overlay of GFP intensity profiles of focused single cells for VCN 0–4 cells. Gating strategy for cell analysis: single cells (area vs. aspect ratio) > focused cells (gradient RMS) > intensity of specific channel.

**Figure 3 ijms-26-05496-f003:**
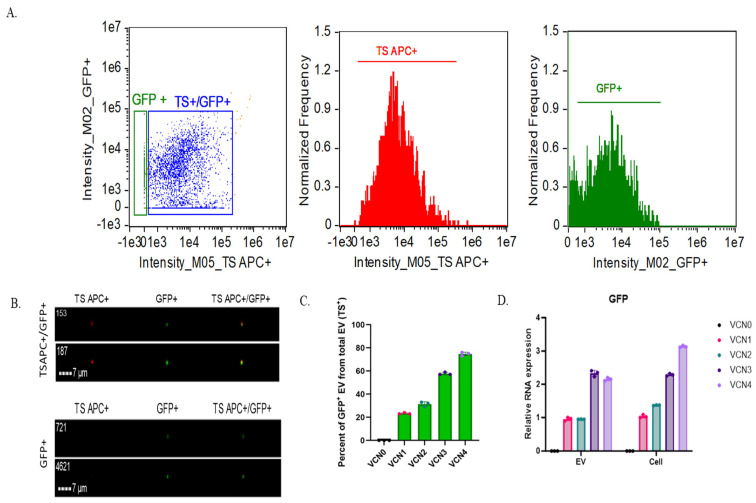
Analysis of VCN EV cargo. (**A**) IFC analysis of VCN4-derived EVs (gating strategy: raw max pixel histogram to separate “SBs” and “not SBs” events > intensity (dot plot showing all “not SB” events (left)) > intensity (histograms displaying events under “TS^+^/GFP^+^” gate (center) and “GFP^+^” gate (right) from “TS^+^/GFP^+^” events). (**B**) Representative images of TSAPC^+^/GFP^+^ EVs (top; particles from Fig. 3A left, blue box) and GFP^+-^only particles (bottom; particles from Fig. 3A left, green box) for VCN 1–4 cells. (**C**) Percent of GFP^+^ EVs of total EVs (TS^+^) for VCN 0–4 cells; total EVs (TS APC^+^) and GFP^+^ EVs were counted for each sample in three extractions after background subtraction (antibody-only counts for TS APC^+^ EVs or buffer-only counts for GFP^+^ EVs), and percentage of GFP^+^ EV was calculated using average for each population from three extractions after background subtraction. Error bars show standard deviation. (**D**) Relative *GFP* RNA expression in VCN 0–4 EVs and cells (color blind friendly scheme: VCN 0 black, VCN 1 pink, VCN 2 green, VCN 3 mauve taupe, and VCN 4 pale violet). Relative *GFP* RNA expression was normalized to VCN1 EVs of cells, respectively. Each bar presents the mean ± standard deviation from three replicates. RNA expression is shown relative to *GAPDH* expression for both EVs and cells.

**Figure 4 ijms-26-05496-f004:**
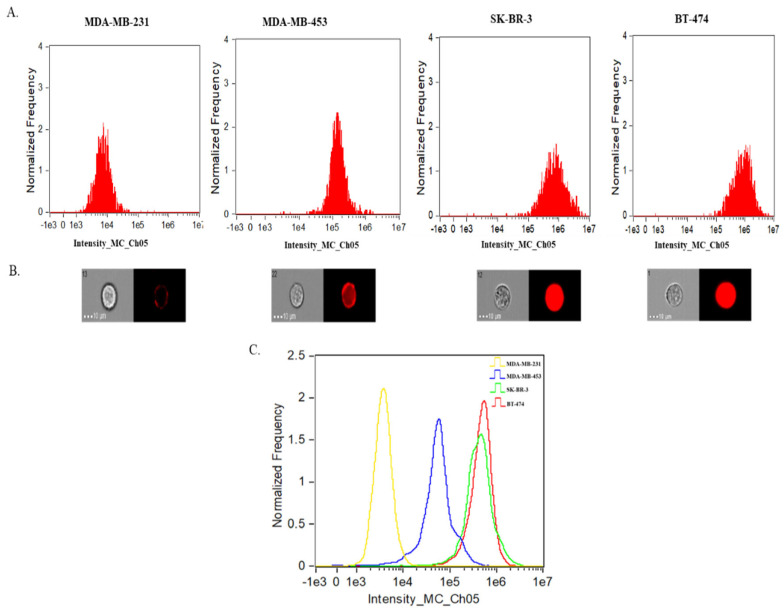
Analysis of SRM 2373 cell lines by IFC. (**A**) HER2- Alexa Fluor^®^ (AF) 647 intensity profile of focused single cells for MDA-MB-231, MDA-MB-453, SK-BR-3, and BT-474 cells (left to right). (**B**) Corresponding representative images of MDA-MB-231, MDA-MB-453, SK-BR-3, and BT-474 cells in the BF channel and HER2 channel after staining with HER2-AF647 antibodies. (**C**) Histogram overlay of (**A**); overlay of HER2-AF647 intensity profiles of focused single cells for MDA-MB-231, MDA-MB-453, SK-BR-3, and BT-474 cells. Gating strategy for cell analysis: single cells (area vs. aspect ratio) > focused cells (gradient RMS) > intensity of specific channel.

**Figure 5 ijms-26-05496-f005:**
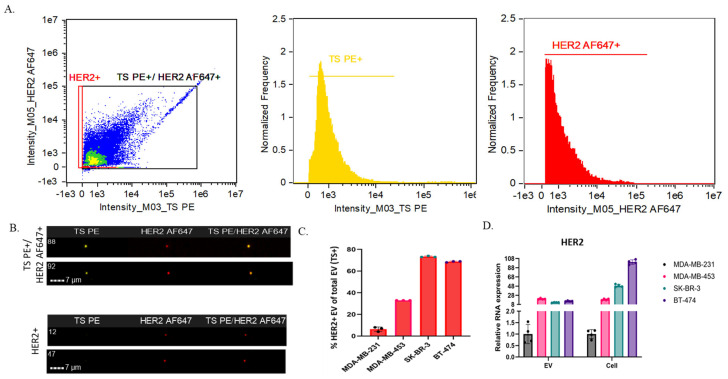
Analysis of SRM 2373 cell EV cargo. (**A**) Representative IFC analysis of BT-474 derived EVs (gating strategy: raw max pixel histogram to separate “SBs” and “not SBs” events > intensity (dot plot showing all “not SB” events (left)) > intensity (histograms displaying events under “TS PE^+^” gate (center) and “HER2AF647^+^” gate (right) from “TSPE^+^/HER2AF647^+^” events). (**B**) Representative images of TS PE^+^/HER2 AF647^+^ EVs (top; particles from Fig. 5A left, black box) or HER2^+^ particles (bottom; particles from Fig. 5A left, red box) from BT-474 cells by IFC. (**C**) Percent of HER2^+^ EV out of total EVs (TS^+^) for MDA-MB-231, MDA-MB-453, SK-BR-3, and BT-474 cells; total EVs (TS PE^+^) and HER2 AF647^+^ EVs were counted for each sample in three replicates after background subtraction (antibody only counts), and percentage of HER2^+^ EVs was calculated by using average for each population from three runs after background subtraction. Error bars show standard deviation. (**D**) Relative *HER2* RNA expression in MDA-MB-231, MDA-MB-453, SK-BR-3, and BT-474 EVs and cells after normalization to MDA-MB-231 relative to *HER2* RNA expression (color blind friendly scheme: MDA-MB-231 black, MDA-MB-453 pink, SK-BR-3 green, and BT-474 mauve taupe). Data are presented as mean ± standard deviation. *HER2* RNA expression is shown relative to *GAPDH* expression for both EVs and cells.

**Table 1 ijms-26-05496-t001:** Summary of particle count of VCN 0–4 cell-derived EVs with Spectradyne nCS1. Data are presented as mean particles (1/mL ± standard deviation) from 3 different extractions.

EV Sample	Particles (1/mL)
VCN 0	5.2 × 10^10^ (±3.3 × 10^10^)
VCN 1	3.8 × 10^10^ (±1.7 × 10^10^)
VCN 2	2.8 × 10^10^ (±2.4 × 10^10^)
VCN 3	4.5 × 10^10^ (±3.4 × 10^10^)
VCN 4	5.7 × 10^10^ (±3.5 × 10^10^)

**Table 2 ijms-26-05496-t002:** Summary of particle count of SRM 2373 cell-derived EVs with Spectradyne nCS1. Data are presented as mean (counts/mL ± standard deviation) from 3 different extractions.

EV Sample	Particles (1/mL)
MDA-MB-231	3.0 × 10^10^ (±1.2 × 10^10^)
MDA-MB-453	9.3 × 10^10^ (±2.5 × 10^9^)
SK-BR-3	1.3 × 10^10^ (±6.7 × 10^9^)
BT-474	3.9 × 10^10^ (±2.3 × 10^10^)

**Table 3 ijms-26-05496-t003:** Summary of Imaging Flow Cytometry analysis of total EVs (TS APC^+^) and GFP^+^ EV counts from three extractions for VCN 0–4 cell-derived EVs. Data are presented as mean particle counts (1/mL) ± standard deviation.

EV Sample	TS^+^ Particles (1/mL)	GFP^+^ Particles (1/mL)	% of GFP^+^ EVs of Total EVs (TS^+^)
VCN0	1.6 × 10^9^ (±1.9 × 10^5^)	4.4 × 10^6^ (±1.4 × 10^3^)	0%
VCN1	2.3 × 10^9^ (±2.0 × 10^5^)	5.3 × 10^8^ (±6.3 × 10^4^)	23%
VCN2	3.7 × 10^9^ (±3.4 × 10^5^)	1.2 × 10^9^ (±7.8 × 10^4^)	31%
VCN3	1.5 × 10^9^ (±7.7 × 10^4^)	8.4 × 10^8^ (±4.0 × 10^4^)	57%
VCN4	2.7 × 10^9^ (±3.4 × 10^5^)	2.00 × 10^9^ (±2.1 × 10^5^)	74%

**Table 4 ijms-26-05496-t004:** Summary of Imaging Flow Cytometry analysis of total EVs (TS PE^+^) and HER2^+^ (AF644) EV counts from three extractions for MDA-MB-231, MDA-MB-453, SK-BR-3, and BT-474 cell-derived EVs. Data are presented as mean particle counts (1/mL) ± standard deviation.

EV Sample	TS^+^ Particles (1/mL)	HER2^+^ Particles (1/mL)	% of HER2^+^ EVs of Total EVs (TS^+^)
MDA-MB-231	3.6 × 10^9^ (±1.9 × 10^6^)	2.1 × 10^8^ (±7.4 × 10^5^)	6%
MDA-MB-453	1.1 × 10^9^ (±6.1 × 10^6^)	3.7 × 10^8^ (±2.2 × 10^6^)	33%
SK-BR-3	1.3 × 10^9^ (±9.0 × 10^5^)	9.2 × 10^8^ (±6.2 × 10^5^)	73%
BT-474	5.6 × 10^9^ (±7.0 × 10^6^)	3.9 × 10^9^ (±4.7 × 10^6^)	69%

## Data Availability

Data is contained within the article and Appendix A.

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
