# Peer review of "Gene Copy Number Dictates Extracellular Vesicle Cargo"

_ijms, 2025, doi:10.3390/ijms26125496_

Round 1

Reviewer 1 Report

Comments and Suggestions for Authors

This study systematically investigated the influence of gene copy number on protein and RNA loads of extracellular vesicles (EVs). By stably transfecting vector copy cells (VCN 0-4) with a fluorescent reporter gene (GFP) and breast cancer cells with varying Her2 copy numbers, demonstrated that: The levels of GFP/Her2 proteins in EVs are positively correlated with their respective gene copy numbers, with a notable threshold effect observed. However, no significant correlation was found between EV RNA load and gene copy number, indicating distinct mechanisms underlying the loading of RNA and proteins into EVs. This research innovatively elucidates the direct regulation of gene copy number on EV protein load, offering novel insights for cancer diagnostics, EV engineering, and drug delivery applications. Nevertheless, several areas warrant further clarification.

Recommendation: minor revision

  1. A detailed explanation is required regarding why droplet digital PCR (ddPCR) might underestimate EV RNA levels (e.g., due to RNA fragmentation), along with suggestions to supplement the analysis with RNA-Seq data or citations of relevant studies.
  2. Further exploration of the potential causes underlying the observed protein threshold effect is recommended.
  3. A deeper investigation into why protein load correlates with gene copy number but not with RNA load is necessary.
  4. The current findings, which focus on two specific genes (GFP and Her2), should be evaluated for generalizability to other genes.
  5. More related work should be cited, e.g. Coord. Chem. Rev.. 2020, 424, 213506; Chem. Commun., 2025,61, 4123-4146

Author Response

This study systematically investigated the influence of gene copy number on protein and RNA loads of extracellular vesicles (EVs). By stably transfecting vector copy cells (VCN 0-4) with a fluorescent reporter gene (GFP) and breast cancer cells with varying Her2 copy numbers, demonstrated that: The levels of GFP/Her2 proteins in EVs are positively correlated with their respective gene copy numbers, with a notable threshold effect observed. However, no significant correlation was found between EV RNA load and gene copy number, indicating distinct mechanisms underlying the loading of RNA and proteins into EVs. This research innovatively elucidates the direct regulation of gene copy number on EV protein load, offering novel insights for cancer diagnostics, EV engineering, and drug delivery applications. Nevertheless, several areas warrant further clarification.

Recommendation: minor revision

  1. A detailed explanation is required regarding why droplet digital PCR (ddPCR) might underestimate EV RNA levels (e.g., due to RNA fragmentation), along with suggestions to supplement the analysis with RNA-Seq data or citations of relevant studies.
  2. Further exploration of the potential causes underlying the observed protein threshold effect is recommended.
  3. A deeper investigation into why protein load correlates with gene copy number but not with RNA load is necessary.
  4. The current findings, which focus on two specific genes (GFP and Her2), should be evaluated for generalizability to other genes.
  5. More related work should be cited, e.g. Coord. Chem. Rev.. 2020, 424, 213506; Chem. Commun., 2025,61, 4123-4146

Please see below for our responses to the comments. We have pointed to the specific line numbers where the changes related to the comments are made. 

1. A detailed explanation is required regarding why droplet digital PCR (ddPCR) might underestimate EV RNA levels (e.g., due to RNA fragmentation), along with suggestions to supplement the analysis with RNA-Seq data or citations of relevant studies.

We added the following (lines 435-441) to address this comment in the Conclusion section. 

  • In general, protein assays are direct and often more robust in low-input samples than RNA-based methods like RT-qPCR or ddPCR, which are more prone to technical variation and amplification biases especially in degraded or low-abundance samples such as EVs. Further studies have also shown that extracellular RNAs, particularly exosomal miRNAs, are often heavily fragmented and low in abundance, resulting in variable detection across platforms and frequent underestimation by ddPCR thus underscoring the importance of RNA-Seq for accurate and unbiased transcriptome profiling [38], [39].

2. Further exploration of the potential causes underlying the observed protein threshold effect is recommended.

  • We agree and acknowledge this notion in lines 408- 411 of the Conclusion section.

3. A deeper investigation into why protein load correlates with gene copy number but not with RNA load is necessary.

We added the following (lines 443-448) to address this comment in the Conclusion section. 

  • Studies have demonstrated that EVs selectively package specific RNAs, such as RNY3, vtRNA, and MIRLET-7, along with associated proteins like YBX1, IGF2BP2, and SRSF1/2, indicating a regulated sorting mechanism distinct from their cellular origin [36], [41]. An integrative multi-omic analysis of breast cancer cells revealed weak correlations between EV protein and RNA content and their corresponding levels in the parental cells [42], suggesting that EVs selectively enrich certain molecules rather than merely mirroring the source cell’s molecular composition.

4. The current findings, which focus on two specific genes (GFP and Her2), should be evaluated for generalizability to other genes.

  • We agree and have reiterated this notion explicitly in lines 407-408 of the Conclusion section.

5. More related work should be cited, e.g. Coord. Chem. Rev.. 2020, 424, 213506; Chem. Commun., 2025,61, 4123-4146

  • We have added relevant new citations when appropriate; see reply to comment 3 as 2 examples.

Reviewer 2 Report

Comments and Suggestions for Authors

In this manuscript, the authors examine the relationship between gene copy number and extracellular vesicle (EV) cargo including the gene transcript (RNA) and protein. Although carefully planned by using cell lines with varying gene copy number of both endogenous (HER2) and reporter (GFP) proteins, the study has several major issues that need addressing.

  • EV purity is the most important aspect of any EV study. In this manuscript, the authors decided to use a very dirty method of EV purification which also sediments major contaminants including proteins and riboprotein complexes. Since the study looks at RNA and protein, it is likely that the observed results were largely contributed by the EV contaminants rather than the EV themselves. Have the authors tried repeating such experiments with EVs purified from a more stringent method (i.e. size exclusion chromatography or affinity-based purification)? The authors should compare the EV/protein ratios from different purification methods to make sure that the EVs are free of background proteins.
  • For RNA analysis, how were the target RNA measured in the EVs? What was the EV input (counts and protein levels) for RNA extraction? Were the target transcript full length or just fragments? What was the internal control for EV qPCR? Are the measured RNAs inside EVs? How were the replicates done? With conditioned media from 3 different passages or from the same passage?
  • Protein analysis lacks a lot of necessary controls especially FMOs to make sure the gating was done correctly. For example, Figures 3 and 5 lack FMOs to make sure that the TS+/GFP+ or TS+/HER2+ gates were drawn correctly. It seems like there is a population of TS+/target- population on the lower right side of the graph. Also, panel C in each of the figures, how did the authors come to such percentage? Please include a representative graph. Figures 2 and 4 should include FMO controls to confirm the levels of GFP and HER2. As proteins can form aggregates, how do the authors avoid the signals from background antibodies and contaminating protein complexes?
  • Minor points: HER2 should be capitalized as this is a human protein.

Author Response

Reviewer comments:

In this manuscript, the authors examine the relationship between gene copy number and extracellular vesicle (EV) cargo including the gene transcript (RNA) and protein. Although carefully planned by using cell lines with varying gene copy number of both endogenous (HER2) and reporter (GFP) proteins, the study has several major issues that need addressing.

  • EV purity is the most important aspect of any EV study. In this manuscript, the authors decided to use a very dirty method of EV purification which also sediments major contaminants including proteins and riboprotein complexes. Since the study looks at RNA and protein, it is likely that the observed results were largely contributed by the EV contaminants rather than the EV themselves. Have the authors tried repeating such experiments with EVs purified from a more stringent method (i.e. size exclusion chromatography or affinity-based purification)? The authors should compare the EV/protein ratios from different purification methods to make sure that the EVs are free of background proteins.
  • For RNA analysis, how were the target RNA measured in the EVs? What was the EV input (counts and protein levels) for RNA extraction? Were the target transcript full length or just fragments? What was the internal control for EV qPCR? Are the measured RNAs inside EVs? How were the replicates done? With conditioned media from 3 different passages or from the same passage?
  • Protein analysis lacks a lot of necessary controls especially FMOs to make sure the gating was done correctly. For example, Figures 3 and 5 lack FMOs to make sure that the TS+/GFP+ or TS+/HER2+ gates were drawn correctly. It seems like there is a population of TS+/target- population on the lower right side of the graph. Also, panel C in each of the figures, how did the authors come to such percentage? Please include a representative graph. Figures 2 and 4 should include FMO controls to confirm the levels of GFP and HER2. As proteins can form aggregates, how do the authors avoid the signals from background antibodies and contaminating protein complexes?
  • Minor points: HER2 should be capitalized as this is a human protein.

Please see below for the responses to the reviewer comments. We have also specifically provided line numbers to the changes as they relate to the comments . 

  • EV purity is the most important aspect of any EV study. In this manuscript, the authors decided to use a very dirty method of EV purification which also sediments major contaminants including proteins and riboprotein complexes. Since the study looks at RNA and protein, it is likely that the observed results were largely contributed by the EV contaminants rather than the EV themselves. Have the authors tried repeating such experiments with EVs purified from a more stringent method (i.e. size exclusion chromatography or affinity-based purification)? The authors should compare the EV/protein ratios from different purification methods to make sure that the EVs are free of background proteins.
  • We agree that the kit-based precipitation methods are generally considered dirty but still yield moderately pure EVs in a high-throughput fashion, which have been used for EV cargo analysis. A quick search of PubMed showed >50 articles that have used the kit-based methods for EV extraction and analysis.

At least from the protein perspective in our study (figures 3 and 5), we disagree that the observed results are contributed by EV contaminants since we gate out particles that are tetraspanin negative (non EVs, debris, contaminants, etc.) and calculate percentages from particles that are tetraspanin positive (EVs). It is possible that the RNA analysis can be affected by the isolation method.

We are planning to do follow-up studies by utilizing some of the other traditional extraction methods to test the hypothesis.    

We also acknowledge that this method of EV isolation needs to be followed up with other extraction methods (in lines 408- 411)  

  • For RNA analysis, how were the target RNA measured in the EVs? What was the EV input (counts and protein levels) for RNA extraction? Were the target transcript full length or just fragments? What was the internal control for EV qPCR? Are the measured RNAs inside EVs? How were the replicates done? With conditioned media from 3 different passages or the same passage?
  • The following method section (lines 357- 377) answers the input, target measurement, and control questions. We have updated the text of the RNA method section to show starting material (lines 358-360), replicate extraction information (lines 362-363), and internal control. GAPDH was used as an internal control for EV and cellular RNA ddPCR. Updated relevant sections are also pasted below.

RNA Extraction

Fifty µL of approximately 1x109 /µL EVs (based on the Spectradyne nCS1 counts) each were used as starting materials for EV RNA extraction, and approximately 2x105 cells were used for cellular RNA extraction. Total cellular RNA was extracted using the QIAGEN RNeasy Plus Mini Kit (Cat # 74134) by following the manufacturer's protocol. EV RNA was extracted by using Invitrogen™ Total Exosome RNA & Protein Isolation Kit (Cat # 4478545) by following the manufacturer's protocol. Three replicate extractions were performed on media from 3 different passages.

Relative RNA quantification by digital droplet polymerase chain reaction (ddPCR)

Total RNA isolated from breast cancer cells and EVs were diluted in RNA storage solution (RSS, ThermoFisher, Cat # AM7000) with 20 ng/µL yeast tRNA (ThermoFisher, Cat # AM7119 ) to a final concentration of 60-150 pg/µL. Diluted total RNA was used for determination of relative GFP (TaqMan™ minor groove binder (MGB) Probe sets [32] or HER2 expression in cells and EVs by using One-Step RT-ddPCR Advanced Kit for Probes (Bio-Rad, Cat # 1864022). HER2-specific primers and probe sequences were published before [19]. GAPDH expression assay (including primers and probe) was purchased from ThermoFisher (Cat # 402869) and was used as an internal control. The ddPCR reactions were set up by following the kit protocols. After the droplets were generated on the Bio-Rad manual droplet generator, they were transferred to a Bio-Rad 96-well ddPCR plate and followed by thermal cycling amplification. The thermal cycling steps were 25°C 3 min for 1 cycle; 50°C 60 min for 1 cycle; 95°C 10 min for 1 cycle; 95°C 30 sec, 58°C 60 sec for 40 cycles; 98°C 10 min for 1 cycle and then store at 4°C. The positive droplets were determined on a Bio-Rad QX200 droplet reader. The relative RNA quantifications were normalized to the GAPDH level of each sample.

Several studies have pointed to the fragmented nature of RNA in EV cargo (also cited in this paper). We expect this to be the case in our RNA ddPCR analysis. We have discussed the limitations of ddPCR analysis at length in terms of EV RNA analysis in the conclusion section. Other methods like RNA seq can be used to investigate the full or fragmented nature of RNA. We intend to collaborate with our partners to study this in the future.

At this point, we are unsure if the RNA is in the EV core or the membrane. A RNase-based assay with ddPCR to determine where the RNA is located is quite difficult, as RNase is something you want to avoid or eliminate when working with RNA in ddPCR. This is because RNases degrade RNA, RNA is the template for reverse transcription (RT) in RT-ddPCR, and even trace amounts of RNase can compromise results. RNase can and has been used for the quantitation of DNA in ddPCR in rare studies.

  • Protein analysis lacks a lot of necessary controls, especially FMOs to make sure the gating was done correctly. For example, Figures 3 and 5 lack FMOs to make sure that the TS+/GFP+ or TS+/HER2+ gates were drawn correctly. It seems like there is a population of TS+/target- population on the lower right side of the graph. Also, panel C in each of the figures, how did the authors come to such percentage? Please include a representative graph. Figures 2 and 4 should include FMO controls to confirm the levels of GFP and HER2. As proteins can form aggregates, how do the authors avoid the signals from background antibodies and contaminating protein complexes?

We have added all the controls figures for figures 3 and 5 in the supplemental information section as supplemental figures 1 and 2 for VCN  and HER2 EV analysis, respectively, with the same gating as VCN 4 EV with TS APC antibody or BT-474 EV with HER2 AF647 + TS PE applied throughout the controls. These are also reflected in lines 147-149 and 224-227.

Panel C graph for figures 3 and 5 comes directly from Table 3 for VCN percentages and Table 4 for SRM 2373 percentages. We are not sure which graph is being requested. Here are example calculations for VCN4 percentage (2.00E+9)/ (2.7E+09) *100= 74%, VCN3 (8.4E+08)/ (1.5E+09) *100 and so on. These percentages are represented as a graph in panel C of Figure 3 (likewise for Figure 5 with Table 4 numbers). This is also detailed in the figure legend for each figure.

Figure 2 shows the cellular GFP status of VCN cell series. These cells express stable GFP, and there is no staining involved. the FMO control for this series is VCN0 cell, which is shown in the figure (Figure 2A (left), 2B (cell image), and 2D (black trendline)). Likewise, in Figure 4, SRM2373 cells are stained with a single antibody (HER2 AF647) to check the cellular protein status. The FMO for this series would be unstained cells, so there would be no signal in this channel with the single stain.

We agree there are protein aggregates that can be mistaken for EVs. To avoid this, we qualitatively analyze the images of the single EV and remove suspected aggregates and other large background noise caused by the debris. In addition, particles that are not positive for tetraspanin (TS) are not (also gated in figures 3 and 5, supplemental figures 1 and 2, dot plots) used for the calculation of positive population percentages. 

  • Minor points: HER2 should be capitalized as this is a human protein.
  • Thank you for bringing this to our attention. All HER2 in the text have been capitalized to reflect human protein.

Round 2

Reviewer 2 Report

Comments and Suggestions for Authors

Thanks for addressing my concerns.